# LS³: Latent Space Safe Sets for Long-Horizon Visuomotor Control of Sparse Reward Iterative Tasks

**Albert Wilcox**[*], **Ashwin Balakrishna**[*], **Brijen Thananjeyan**,
**Joseph E. Gonzalez**, **Ken Goldberg**

\* equal contribution

{albertwilcox, ashwin_balakrishna}@berkeley.edu

**Abstract:** Reinforcement learning (RL) has shown impressive success in exploring high-dimensional environments to learn complex tasks, but can often exhibit unsafe behaviors and require extensive environment interaction when exploration is unconstrained. A promising strategy for learning in dynamically uncertain environments is requiring that the agent can robustly return to learned safe sets, where task success (and therefore safety) can be guaranteed. While this approach has been successful in low-dimensions, enforcing this constraint in environments with visual observations is exceedingly challenging. We present a novel continuous representation for safe sets by framing it as *a binary classification problem* in a learned latent space, which flexibly scales to image observations. We then present a new algorithm, Latent Space Safe Sets (LS³), which uses this representation for long-horizon tasks with sparse rewards. We evaluate LS³ on 4 domains, including a challenging sequential pushing task in simulation and a physical cable routing task. We find that LS³ can use prior task successes to restrict exploration and learn more efficiently than prior algorithms while satisfying constraints. See https://tinyurl.com/latent-ss for code and supplementary material.

**Keywords:** Reinforcement Learning, Imitation Learning, Safety

## 1 Introduction

Visual planning over learned forward dynamics models is a popular area of research in robotic control from images [1, 2, 3, 4, 5, 6, 7], as it enables closed-loop, model-based control for tasks where the state of the system is not directly observable or difficult to analytically model, such as the configuration of a sheet of fabric or segment of cable. These methods learn predictive models over either images or a learned latent space, which can then be used to plan actions which maximize some task reward. While these approaches have significant promise, there are several open challenges in learning policies from visual observations. First, reward specification is particularly challenging for visuomotor control tasks, because high-dimensional observations often do not expose the necessary features required to design dense, informative reward functions [8], especially for long-horizon tasks. Second, while many prior reinforcement learning methods have been successfully applied to image-based control tasks [9, 10, 11, 12, 13], learning policies from image observations often requires extensive exploration due to the high dimensionality of the observation space and the difficulties in reward specification, making safe and efficient learning exceedingly challenging.

One promising strategy for efficiently learning safe control policies is to learn a safe set [14, 15], which captures the set of states from which the agent is known to behave safely, which is often reformulated as the set of states where it has previously completed the task. When used to restrict exploration, this safe set can be used to enable highly efficient and safe learning [14, 16, 17], as exploration is restricted to states in which the agent is confident in task success. However, while these safe sets can give rise to algorithms with a number of appealing theoretical properties such as convergence to a goal set, constraint satisfaction, and iterative improvement [14, 16, 18], using them for controller design for practical problems requires developing continuous approximations at the expense of maintaining theoretical guarantees [17]. This choice of continuous approximation is a key element in determining the applications to which these safe sets can be used for control.

University of California, Berkeley.

5th Conference on Robot Learning (CoRL 2021), London, UK.

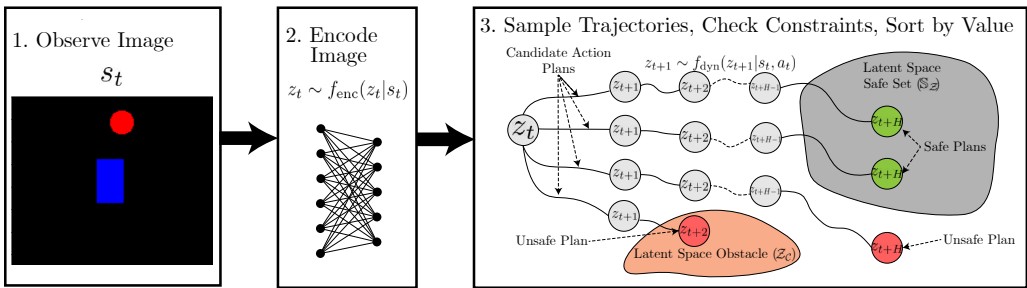

Figure 1: **Latent Space Safe Sets (LS$^3$):** At time $t$, LS$^3$ observes an image $s_t$ of the environment. The image is first encoded to a latent vector $z_t \sim f_{\text{enc}}(z_t|s_t)$. Then, LS$^3$ uses a sampling-based optimization procedure to optimize $H$-length action sequences by sampling $H$-length latent trajectories over the learned latent dynamics model $f_{\text{dyn}}$. For each sampled trajectory, LS$^3$ checks whether latent space obstacles are avoided and if the terminal state in the trajectory falls in the latent space safe set. The terminal state constraint encourages the algorithm to maintain plans back to regions of safety and task confidence, but still enables exploration. For feasible trajectories, the sum of rewards and value of the terminal state are computed and used for sorting. LS$^3$ executes the first action in the optimized plan and then performs this procedure again at the next timestep.

Prior works have presented approaches which collect a discrete safe set of states from previously successful trajectories and represent a continuous relaxation of this set by constructing a convex hull of these states [14] or via kernel density estimation with a tophat kernel function [17]. While these approaches have been successful for control tasks with low-dimensional states, extending them to high-dimensional observations presents two key challenges: (1) *scalability:* these prior methods cannot be efficiently applied when the number of observations in prior successful trajectories is large, as querying safe set inclusion scales linearly with number of samples it contains and (2) *representation capacity:* both of these prior approaches do not scale well to high dimensional observations and are limited in the space of continuous sets that they can efficiently represent. Applying these ideas to visuomotor control is even more challenging, since images do not directly expose details about the system state or dynamics that are typically needed for formal controller analysis [14, 16, 19].

This work makes several contributions. First, we introduce a scalable continuous approximation method which makes it possible to leverage safe sets for visuomotor policy learning. The key idea is to reframe the safe set approximation as a *binary classification problem* in a learned latent space, where the objective is to distinguish states from successful trajectories from those in unsuccessful trajectories. Second, we present Latent Space Safe Sets (LS$^3$), a model-based RL algorithm which encourages the agent to maintain plans back to regions in which it is confident in task completion, even when learning in high dimensional spaces. This constraint makes it possible to define a control strategy to (1) improve safely by encouraging consistent task completion (and therefore avoid unsafe behavior) and (2) learn efficiently since the agent only explores promising states in the immediate neighborhood of those in which it was previously successful. Third, we present simulation experiments on 3 visuomotor control tasks which suggest that LS$^3$ can learn to improve upon demonstrations more safely and efficiently than prior algorithms. Fourth, we conduct physical experiments on a vision-based cable routing task which suggest that LS$^3$ can learn more efficiently than prior algorithms while consistently completing the task and satisfying constraints during learning.

## 2 Related Work

### 2.1 Safe, Iterative Learning Control

In iterative learning control (ILC), the agent tracks a reference trajectory and uses data from controller rollouts to refine tracking performance [20]. Rosolia et al. [21], Rosolia and Borrelli [18, 14] present a new class of algorithms, known as Learning Model Predictive Control (LMPC), which are reference-free and instead iteratively *improve* upon the performance of an initial feasible trajectory. To achieve this, Rosolia et al. [21], Rosolia and Borrelli [18, 14] use data from controller rollouts to learn a safe set and value function, with which recursive feasibility, stability, and local optimality can be guaranteed given a known, deterministic nonlinear system or stochastic linear system under certain regularity assumptions. However, a core challenge with these algorithms is that they assume known system dynamics, and cannot be applied to high-dimensional control problems. Thananjeyan et al. [17] extends the LMPC framework to higher dimensional settings in which system dynamics are unknown and must be learned, but the visuomotor control setting introduces a number of new

challenges as learned system dynamics, safe sets, and value functions must flexibly scale to visual inputs. Richards et al. [15] designs expressive safe sets for fixed policies using neural network classifiers with Lyapunov constraints. In contrast, LS$^3$ constructs a safe set for an improving policy by optimizing a task cost function instead of uniformly expanding across the state space.

## 2.2 Model Based Reinforcement Learning

There has been significant recent progress in algorithms which combine ideas from model-based planning and control with deep learning [22, 23, 24, 25, 26, 27]. These algorithms are gaining popularity in the robotics community as they enable leaning complex policies from data while maintaining some of the sample efficiency and safety benefits of classical model-based control techniques. However, these algorithms typically require hand-engineered dense cost functions for task specification, which can often be difficult to provide, especially in high-dimensional spaces. This motivates leveraging demonstrations (possibly suboptimal) to provide an initial signal regarding desirable agent behavior. There has been some prior work on leveraging demonstrations in model-based algorithms such as Quinlan and Khatib [28] and Ichnowski et al. [29], which use model-based control with known dynamics to refine initially suboptimal motion plans, and Fu et al. [24], which uses demonstrations to seed a learned dynamics model for fast online adaptation using iLQR [24]. Thananjeyan et al. [17], Zhu et al. [30] present ILC algorithms which rapidly improve upon suboptimal demonstrations when system dynamics are unknown. However, these algorithms either require knowledge of system dynamics [28, 29] or are limited to low-dimensional state spaces [24, 17, 30] and cannot be flexibly applied to visuomotor control tasks.

## 2.3 Reinforcement Learning from Pixels

Reinforcement learning and model-based planning from visual observations is gaining significant recent interest as RGB images provide an easily available observation space for robot learning [1, 31]. Recent work has proposed a number of model-free and model-based algorithms that have seen success in laboratory settings in a number of robotic tasks when learning from visual observations [32, 33, 10, 34, 12, 13, 1, 35, 31]. However, two core issues that prevent application of many RL algorithms in practice, inefficient exploration and safety, are significantly exacerbated when learning from high-dimensional visual observations in which the space of possible behaviors is very large and the features required to determine whether the robot is safe are not readily exposed. There has been significant prior work on addressing inefficiencies in exploration for visuomotor control such as latent space planning [2, 31, 35] and goal-conditioned reinforcement learning [13, 10]. However, safe reinforcement learning for visuomotor tasks has received substantially less attention. Thananjeyan et al. [36] and Kahn et al. [37] present reinforcement learning algorithms which estimate the likelihood of constraint violations to avoid them [36] or reduce the robot's velocity [37]. Unlike these algorithms, which focus on presenting methods to avoid violating user-specified constraints, LS$^3$ additionally provides consistent task completion during learning by limiting exploration to the neighborhood of prior task successes. This difference makes LS$^3$ less susceptible to the challenges of unconstrained exploration present in standard model-free reinforcement learning algorithms.

## 3 Problem Statement

We consider an agent interacting in a finite horizon goal-conditioned Markov Decision Processes (MDP) which can be described with the tuple $\mathcal{M} = (\mathcal{S}, \mathcal{G}, \mathcal{A}, P(\cdot|\cdot, \cdot), R(\cdot, \cdot), \mu, T)$. $\mathcal{S}$ and $\mathcal{A}$ are the state and action spaces, $P : \mathcal{S} \times \mathcal{A} \times \mathcal{S} \to [0, 1]$ maps a state and action to a probability distribution over subsequent states, $R : \mathcal{S} \times \mathcal{A} \times \mathcal{S} \to \mathbb{R}$ is the reward function, $\mu$ is the initial state distribution ($s_0 \sim \mu$), and $T$ is the time horizon. In this work, the agent is only provided with RGB image observations $s_t \in \mathbb{R}_+^{W \times H \times 3} = \mathcal{S}$, where $W$ and $H$ are the image width and height in pixels, respectively. We consider iterative tasks, where the agent must reach a fixed goal set $\mathcal{G} \subseteq \mathcal{S}$ as efficiently as possible and the support of $\mu$ is small. While there are a number of possible choices of reward functions that would encourage fast convergence to $\mathcal{G}$, providing shaped reward functions can be exceedingly challenging, especially when learning from high dimensional observations. Thus, as in Thananjeyan et al. [17], we consider a sparse reward function that only indicates task completion: $R(s, a, s') = 0$ if $s' \in \mathcal{G}$ and $-1$ otherwise. To incorporate constraints, we augment $\mathcal{M}$ with an extra constraint indicator function $\mathcal{C} : \mathcal{S} \to \{0, 1\}$ which indicates whether a state satisfies user-specified state-space constraints, such as avoiding known obstacles. This is consistent with the modified CMDP formulation used in [36]. We assume that $R$ and $\mathcal{C}$ can be evaluated on the current state of the system, but may be approximated using prior data for use during planning. We make this assumption because

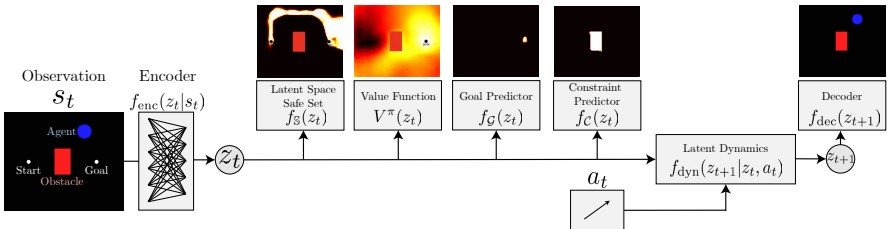

Figure 2: **LS$^3$ Learned Models**: LS$^3$ learns a low-dimensional latent representation of image-observations (Section 4.1) and learns a dynamics model, value function, reward function, constraint classifier, and safe set for constrained planning and task-completion driven exploration in this learned latent space. These models are then used for model-based planning to maximize the total value of predicted latent states (Section 4.3) while enforcing the safe set (Section 4.2) and user-specified constraints (Section 4.3).

in practice we plan over predicted future states, which may not be predicted at sufficiently high fidelity to expose the necessary information to directly evaluate $R$ and $C$ during planning.

Given a policy $\pi : \mathcal{S} \to \mathcal{A}$, we define its expected total return in $\mathcal{M}$ as $R^\pi = \mathbb{E}_{\pi,\mu,P} \left[ \sum_t R(s_t, a_t) \right]$. Furthermore, we define $P_C^\pi(s)$ as the probability of future constraint violation (within time horizon $T$) under policy $\pi$ from state $s$. The objective is to maximize the expected return $R^\pi$ while maintaining a constraint violation probability lower than $\delta_\mathcal{C}$. This can be written formally as follows:

$$\pi^* = \arg\max_{\pi \in \Pi} \{ R^\pi : \mathbb{E}_{s_0 \sim \mu} \left[ P_C^\pi(s_0) \right] \leq \delta_\mathcal{C} \} \qquad (1)$$

We assume that the agent is provided with an offline dataset $\mathcal{D}$ of transitions in the environment of which some subset $\mathcal{D}_{\text{constraint}} \subsetneq \mathcal{D}$ are constraint violating and some subset $\mathcal{D}_{\text{success}} \subsetneq \mathcal{D}$ appear in successful demonstrations from a suboptimal supervisor. As in [36], $\mathcal{D}_{\text{constraint}}$ contains examples of constraint violating behaviors (for example from prior runs of different policies or collected under human supervision) so that the agent can learn about states which violate user-specified constraints.

# 4 Latent Space Safe Sets (LS$^3$)

We describe how LS$^3$ uses demonstrations and online interaction to safely learn iteratively improving policies. Section 4.1 describes how we learn a low-dimensional latent representation of image observations to facilitate efficient model-based planning. To enable this planning, we learn a probabilistic forward dynamics model as in [26] in the learned latent space and models to estimate whether plans will likely complete the task (Section 4.2) and to estimate future rewards and constraint violations (Section 4.3) from predicted trajectories. In Section 4.4, we discuss how these components are synthesized in LS$^3$. Dataset $\mathcal{D}$ is expanded using online rollouts of LS$^3$ and used to update all latent space models (Sections 4.2 and 4.3) after every $K$ rollouts. See Algorithm 1 and the supplement for further details on training procedures and data collection.

---

**Algorithm 1** Latent Space Safe Sets (LS$^3$)

---

**Require:** offline dataset $\mathcal{D}$, number of updates $U$
 1: Train VAE encoder $f_{\text{enc}}$ and decoder $f_{\text{dec}}$ (Section 4.1) using data from $\mathcal{D}$
 2: Train dynamics $f_{\text{dyn}}$, safe set classifier $f_\mathbb{S}$(Section 4.2), and the value function $V$ goal indicator $f_\mathcal{G}$, and constraint estimator $f_\mathcal{C}$ (Section 4.3) using data from $\mathcal{D}$.
 3: **for** $j \in \{1, \ldots, U\}$ **do**
 4:     **for** $k \in \{1, \ldots, K\}$ **do**
 5:         Sample starting state $s_0$ from $\mu$.
 6:         **for** $t \in \{1, \ldots, T\}$ **do**
 7:             Choose and execute $a_t$ (Section 4.4)
 8:             Observe $s_{t+1}$, reward $r_t$, constraint $c_t$.
 9:             $\mathcal{D} := \mathcal{D} \cup \{ (s_t, a_t, s_{t+1}, r_t, c_t) \}$
10:     Update $f_{\text{dyn}}$, $V$, $f_\mathcal{G}$, $f_\mathcal{C}$, and $f_\mathbb{S}$ with data from $\mathcal{D}$.

---

## 4.1 Learning a Latent Space for Planning

Learning compressed representations of images has been a popular approach in vision based control to facilitate efficient algorithms for planning and control which can reason about lower dimensional

inputs [2, 35, 6, 38, 39, 31]. To learn such a representation, we train a $\beta$-variational autoencoder [40] on states in $\mathcal{D}$ to map states to a probability distribution over a $d$-dimensional latent space $\mathcal{Z}$. The resulting encoder network $f_{\text{enc}}(z|s)$ is then used to sample latent vectors $z_t \sim f_{\text{enc}}(z_t|s_t)$ to train a forward dynamics model, value function, reward estimator, constraint classifier, safe set, and combine these elements to define a policy for model-based planning. Motivated by Laskin et al. [41], during training we augment inputs to the encoder with random cropping, which we found to be helpful in learning representations that are useful for planning. For all environments we use a latent dimension of $d = 32$, as in [2] and found that varying $d$ did not significantly affect performance.

## 4.2 Latent Safe Sets for Model-Based Control

LS$^3$ learns a binary classifier for latent states to learn a latent space safe set that represents states from which the agent has high confidence in task completion based on prior experience. Because the agent can reach the goal from these states, they are safe: the agent can avoid constraint violations by simply completing the task as before. While classical algorithms use known dynamics to construct safe sets, we approximate this set using successful trajectories from prior iterations. At each iteration $j$, the algorithm collects $K$ trajectories in the environment. We then define the sampled safe set at iteration $j$, $\mathbb{S}^j$, as the set of states from which the agent has successfully navigated to $\mathcal{G}$ in iterations $0$ through $j$ of training, where demonstrations trajectories are those collected at iteration 0. We refer to the dataset collecting all these states as $\mathcal{D}_{\text{success}}$. This discrete set is difficult to plan to with continuous-valued state distributions so we leverage data from $\mathcal{D}_{\text{success}}$ (data in the sampled safe set), data from $\mathcal{D} \setminus \mathcal{D}_{\text{success}}$ (data outside the sampled safe set), and the learned encoder from Section 4.1 to learn a continuous relaxation of this set in latent space (the latent safe set). We train a neural network with a binary cross-entropy loss to learn a binary classifier $f_{\mathbb{S}}(\cdot)$ that predicts the probability of a state $s_t$ with encoding $z_t$ being in $\mathbb{S}^j$. To mitigate the negative bias that appears when trajectories that start in safe regions fail, we utilize the intuition that if a state $s_{t+1} \in \mathbb{S}^j$ then it is likely that $s_t$ is also safe. To do this, rather than just predict $\mathbb{1}_{\mathbb{S}^j}$, we train $f_{\mathbb{S}}$ with a recursive objective to predict $\max(\mathbb{1}_{\mathbb{S}^j}, \gamma_{\mathbb{S}} f_{\mathbb{S}}(s_{t+1}))$. The relaxed latent safe set is parameterized by the superlevel sets of $f_{\mathbb{S}}$, where the level $\delta_{\mathbb{S}}$ is adaptively set during execution: $\mathbb{S}^j_{\mathcal{Z}} = \{z_t | f_{\mathbb{S}}(\cdot)(z_t) \geq \delta_{\mathbb{S}}\}$.

## 4.3 Reward and Constraint Estimation

In this work, we define rewards based on whether the agent has reached a state $s \in \mathcal{G}$, but we need rewards that are defined on predictions from the dynamics, which may not correspond to valid real images. To address this, we train a classifier $f_{\mathcal{G}} : \mathcal{Z} \to \{0, 1\}$ to map the encoding of a state to whether the state is contained in $\mathcal{G}$ using terminal states in $\mathcal{D}_{\text{success}}$ (which are known to be in $\mathcal{G}$) and other states in $\mathcal{D}$. However, in the temporally-extended, sparse reward tasks we consider, reward prediction alone is insufficient because rewards only indicate whether the agent is in the goal set, and thus provide no signal on task progress unless the agent can plan to the goal set. To address this, as in prior MPC-literature [17, 16, 14, 8], we train a recursively-defined value function (details in the supplement). Similar to the reward function, we use the encoder (Section 4.1) to train a classifier $f_{\mathcal{C}} : \mathcal{Z} \to [0, 1]$ with data of constraint violating states from $\mathcal{D}_{\text{constraint}}$ and the constraint satisfying states in $\mathcal{D} \setminus \mathcal{D}_{\text{constraint}}$ to map the encoding of a state to the probability of constraint violation.

## 4.4 Model-Based Planning with LS$^3$

LS$^3$ aims to maximize total rewards attained in the environment while limiting constraint violation probability within some threshold $\delta_{\mathcal{C}}$ (equation 1). We optimize an approximation of this objective over an $H$-step receding horizon with model-predictive control. Precisely, LS$^3$ solves the following optimization problem to generate an action to execute at timestep $t$:

$$\underset{a_{t:t+H-1} \in \mathcal{A}^H}{\arg\max} \quad \mathbb{E}_{z_{t:t+H}} \left[ \sum_{i=1}^{H-1} f_{\mathcal{G}}(z_{t+i}) + V^{\pi}(z_{t+H}) \right] \tag{2}$$

$$\text{s.t.} \quad z_t \sim f_{\text{enc}}(z_t|s_t) \tag{3}$$

$$z_{k+1} \sim f_{\text{dyn}}(z_{k+1}|z_k, a_k) \; \forall k \in \{t, \ldots, t+H-1\} \tag{4}$$

$$\hat{\mathbb{P}}\left(z_{t+H} \in \mathbb{S}^{j-1}_{\mathcal{Z}}\right) \geq 1 - \delta_{\mathbb{S}} \tag{5}$$

$$\hat{\mathbb{P}}(z_{t+i} \in \mathcal{Z}_{\mathcal{C}}) \leq \delta_{\mathcal{C}} \; \forall i \in \{0, \ldots, H-1\} \tag{6}$$

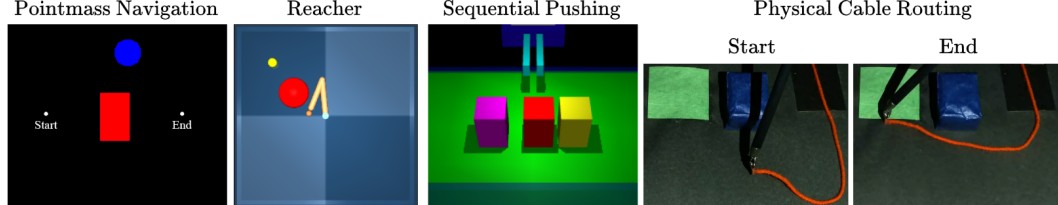

Figure 3: **Experimental Domains:** LS$^3$ is evaluated on 3 long-horizon, image-based, simulation environments: a visual navigation domain where the goal is to navigate the blue point mass to the right goal set while avoiding the red obstacle, a 2 degree of freedom reacher arm where the task is to navigate around a red obstacle to reach the yellow goal set, and a sequential pushing task where the robot must push each of 3 blocks forward a target displacement from left to right. We also evaluate LS$^3$ on a physical, cable-routing task on a da Vinci Surgical Robot, where the goal is to guide a red cable to a green target without the cable or robot arm colliding with the blue obstacle. This requires learning visual dynamics, because the agent must model how the rest of the cable will deform during manipulation to avoid collisions with the obstacle.

In this problem, the expectations and probabilities are taken with respect to the learned, probabilistic dynamics model $f_{\text{dyn}}(z_{t+1}|z_t, a_t)$. The optimization problem is solved approximately using the cross-entropy method (CEM) [42] which is a popular optimizer in model-based RL [43, 17, 16, 44, 36].

The objective function is the expected sum of future rewards if the agent executes $a_{t:t+H-1}$ and then subsequently executes $\pi$ (equation 2). First, the current state $s_t$ is encoded to $z_t$ (equation 3). Then, for a candidate sequence of actions $a_{t:t+H-1}$, an $H$-step latent trajectory $\{z_{t+1}, \ldots, z_{t+H}\}$ is sampled from the learned dynamics $f_{\text{dyn}}$ (equation 4). LS$^3$ constrains exploration using two chance constraints: (1) the terminal latent state in the plan must fall in the safe set (equation 5) and (2) all latent states in the trajectory must satisfy user-specified state-space constraints (equation 6). $\mathcal{Z_C}$ is the set of all latent states such that the corresponding observation is constraint violating. The optimizer estimates constraint satisfaction probabilities for a candidate action sequence by simulating it repeatedly over $f_{\text{dyn}}$. The first chance constraint ensures the agent maintains the ability to return to safe states where it knows how to do the task within $H$ steps if necessary. Because the agent replans at each timestep, the agent need not return to the safe set: during training, the safe set expands, enabling further exploration. In practice, we set $\delta_{\mathbb{S}}$ for the safe set classifier $f_{\mathcal{S}}$ adaptively as described in the supplement. The second chance constraint encourages constraint violation probability of no more than $\delta_{\mathcal{C}}$. After solving the optimization problem, the agent executes the first action in the plan: $\pi(z_t) = a_t$ where $a_t$ is the first element of $a^*_{t:t+H-1}$, observes a new state, and replans.

## 5 Experiments

We evaluate LS$^3$ on 3 robotic control tasks in simulation and a physical cable routing task on the da Vinci Research Kit (dVRK) [45]. Safe RL is of particular interest for surgical robots such as the dVRK due to its delicate structure, motivating safety, and relatively imprecise controls [17, 46], motivating closed-loop control. We study whether LS$^3$ can learn more safely and efficiently than algorithms that do not structure exploration based on prior task successes.

### 5.1 Comparisons

We evaluate LS$^3$ in comparison to prior algorithms that behavior clone suboptimal demonstrations before exploring online (**SACfD**) [47] or leverage offline reinforcement learning to learn a policy using all offline data before updating the policy online (**AWAC**) [48]. For both of these comparisons we enforce constraints via a tuned reward penalty of $\lambda$ for constraint violations as in [49]. We also implement a version of SACfD with a learned recovery policy (**SACfD+RRL**) using the Recovery RL algorithm [36] to use prior constraint violating data to try to avoid constraint violating states. We then compare LS$^3$ to an ablated version without the safe set constraint (just binary classification (BC)) in equation 5 (**LS$^3$ ($-$Safe Set)**) to evaluate if the safe set promotes consistent task completion and stable learning. Finally, we compare LS$^3$ to an ablated version of the safe set classifier (Section 4.2) without a recursive objective, where the classifier is just trained to predict $\mathbb{1}_{\mathbb{S}^j}$ (**LS$^3$ (BC SS)**). See the supplement for details on hyperparameters and offline data used for LS$^3$ and prior algorithms.

### 5.2 Evaluation Metrics

For each algorithm on each domain, we aggregate statistics over random seeds (10 for simulation experiments, 3 for the physical experiment), reporting the mean and standard error across the seeds.

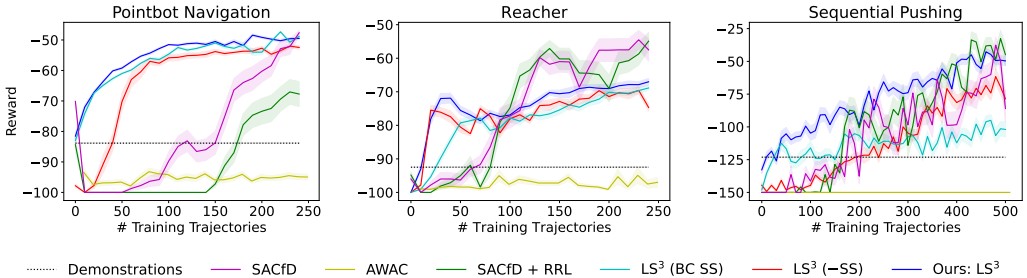

Figure 4: **Simulation Experiments Results:** Learning curves showing mean and standard error over 10 random seeds. We see that LS³ learns more quickly than baselines and ablations. Although SACfD and SACfD+RRL converge to similar reward values, LS³ is much more sample efficient and stable across random seeds.

We present learning curves that show the total sum reward for each training trajectory to study how efficiently LS³ and the comparisons learn each task. Because all tasks use the sparse task completion based rewards defined in Section 3, the total reward for a trajectory is the time to reach the goal set, where more negative rewards correspond to slower convergence to $\mathcal{G}$. Thus, for a task with task horizon $T$, a total reward greater than $-T$ implies successful task completion. The state is frozen in place upon constraint violation until the task horizon elapses. We also report task success and constraint satisfaction rates for LS³ and comparisons during learning to study (1) the degree to which task completion influences sample efficiency and (2) how safely different algorithms explore. LS³ collects $K = 10$ trajectories in between training phases on simulated tasks and $K = 5$ in between training phases for physical tasks, while the SACfD and AWAC comparisons update their parameters after each timestep. This presents a metric in terms of the amount of data collected across algorithms.

## 5.3 Domains

In simulation, we evaluate LS³ on 3 vision-based continuous control domains that are illustrated in Figure 3. We evaluate LS³ and comparisons on a constrained visual navigation task (Pointmass Navigation) where the agent navigates from a fixed start state to a fixed goal set while avoiding a large central obstacle. We study this domain to gain intuition and visualize the learned value function, goal/constraint indicators, and safe set in Figure 2. We then study a constrained image-based reaching task (Reacher) based on [50], where the objective is to navigate the end effector of a 2-link planar robotic arm to a yellow goal position without the end-effector entering a red stay out zone. We then study a challenging sequential image-based robotic pushing domain (Sequential Pushing), in which the objective is to push each of the 3 blocks forward on the table without pushing them to either side and causing them to fall out of the workspace. Finally, we evaluate LS³ with an image-based physical experiment on the da Vinci Research Kit (dVRK) [51] (Figure 3), where the objective is to guide the endpoint of a cable to a goal region without letting the cable or end effector collide with an obstacle. The Pointmass Navigation and Reaching domains have a task horizon of $T = 100$ while the Sequential Pushing domain and physical experiment have task horizons of $T = 150$ and $T = 50$ respectively. See the supplement for more details on all domains.

## 5.4 Simulation Results

We find that LS³ is able to learn more stably and efficiently than all comparisons across all simulated domains while converging to similar performance within 250 trajectories collected online (Figure 4). LS³ is able to consistently complete the task during learning, while the comparisons, which do not learn a safe set to structure exploration based on prior successes, exhibit much less stable learning. Additionally, in Table 1 and Table 2, we report the task success rate and constraint violation rate of all algorithms during training. We find that LS³ achieves a significantly higher task success rate than comparisons on all tasks. We also find that LS³ violates constraints less often than comparisons on the Reacher task, but violates constraints more often than SACfD and SACfD+RRL on the other domains. This is because SACfD and SACfD+RRL spend much less time in the neighborhood of constraint violating states during training due to their lower task success rates. Because they do not efficiently learn to perform the tasks, they do not violate constraints as often. We find that the AWAC comparison achieves very low task performance. While AWAC is designed for offline reinforcement learning, to the best of our knowledge, it has not been previously evaluated on long-horizon, image-based tasks as in this paper, which we hypothesize are very challenging for it.

Table 1: **Task Success Rate over all Training Episodes:** We present the mean and standard error of training-time task completion rate over 10 random seeds. We find LS$^3$ outperforms all comparisons across all 3 domains, with the gap increasing for the challenging sequential pushing task.

|  | SACFD | AWAC | SACFD+RRL | LS$^3$ ($-$SS) | LS$^3$ |
|---|---|---|---|---|---|
| POINTMASS NAVIGATION | $0.363 \pm 0.068$ | $0.312 \pm 0.093$ | $0.184 \pm 0.053$ | $0.818 \pm 0.019$ | $\mathbf{0.988 \pm 0.004}$ |
| REACHER | $0.502 \pm 0.072$ | $0.255 \pm 0.089$ | $0.473 \pm 0.056$ | $0.736 \pm 0.025$ | $\mathbf{0.870 \pm 0.024}$ |
| SEQUENTIAL PUSHING | $0.425 \pm 0.064$ | $0.006 \pm 0.003$ | $0.466 \pm 0.065$ | $0.366 \pm 0.030$ | $\mathbf{0.648 \pm 0.049}$ |

Table 2: **Constraint Violation Rate:** We report mean and standard error of training-time constraint violation rate over 10 random seeds. LS$^3$ violates constraints less than comparisons on the Reacher task, but SAC and SACfD+RRL achieve lower constraint violation rates on the Navigation and Pushing tasks, likely due to spending less time in the neighborhood of constraint violating regions due to their much lower task success rates.

|  | SACFD | AWAC | SACFD+RRL | LS$^3$ ($-$SS) | LS$^3$ |
|---|---|---|---|---|---|
| POINTMASS NAVIGATION | $0.006 \pm 0.002$ | $0.104 \pm 0.070$ | $\mathbf{0.001 \pm 0.001}$ | $0.019 \pm 0.006$ | $0.005 \pm 0.001$ |
| REACHER | $0.146 \pm 0.039$ | $0.398 \pm 0.107$ | $0.142 \pm 0.031$ | $0.247 \pm 0.027$ | $\mathbf{0.102 \pm 0.027}$ |
| SEQUENTIAL PUSHING | $\mathbf{0.033 \pm 0.003}$ | $0.138 \pm 0.028$ | $0.054 \pm 0.006$ | $0.122 \pm 0.031$ | $0.107 \pm 0.016$ |

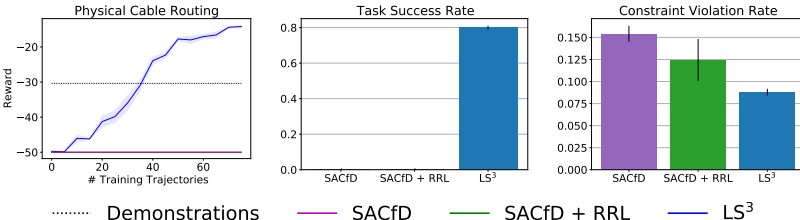

Figure 5: **Physical Cable Routing Results:** We present learning curves, task success rates and constraint violation rates with a mean and standard error across 3 random seeds. LS$^3$ learns a more efficient policy than the demonstrator while still violating constraints less than comparisons, which are unable to learn the task.

We find LS$^3$ has a lower success rate when the safe set constraint is removed (LS$^3$($-$Safe Set)) as expected. The safe set is particularly important in the sequential pushing task, and LS$^3$ ($-$Safe Set) has a much lower task completion rate than LS$^3$. LS$^3$ without the recursive classification objective from Section 4.2 (LS$^3$ (BC SS)) has similar performance to LS$^3$ on the navigation environment, but learns substantially more slowly on the Reacher environment and performs significantly worse than LS$^3$ on the more challenging Pushing environment as the learned safe set is unable to exploit temporal structure to distinguish safe states from unsafe states. See the supplement for details on experimental parameters and offline data used for LS$^3$ and comparisons and ablations studying the effect of the planning horizon and threshold used to define the safe set.

### 5.5 Physical Results

In physical experiments, we compare LS$^3$ to SACfD and SACfD+RRL (Figure 5) on the physical cable routing task illustrated in Figure 3. We find LS$^3$ quickly outperforms the suboptimal demonstrations while succeeding at the task significantly more often than both comparisons, which are unable to learn the task and also violate constraints more than LS$^3$. We hypothesize that the difficulty of reasoning about cable collisions and deformation from images makes it challenging for prior algorithms to make sufficient task progress as they do not use prior successes to structure exploration. See the supplement for details on experimental parameters and offline data used for LS$^3$ and comparisons.

## 6 Discussion and Future Work

We present LS$^3$, a scalable algorithm for safe and efficient policy learning for visuomotor tasks. LS$^3$ structures exploration by learning a safe set in a learned latent space, which captures the set of states from which the agent is confident in task completion. LS$^3$ then ensures that the agent can plan back to states in the safe set, encouraging consistent task completion during learning. Experiments suggest that LS$^3$ can safely and efficiently learn 4 visuomotor control tasks, including a challenging sequential pushing task in simulation and a cable routing task on a physical robot. In future work, we are excited to explore further physical evaluation of LS$^3$ on safety critical visuomotor control tasks and applications to systems with dynamic constraints on velocity or acceleration.

**Acknowledgments**

This research was performed at the AUTOLAB at UC Berkeley in affiliation with the Berkeley AI Research (BAIR) Lab, the Real-Time Intelligent Secure Execution (RISE) Lab, and the CITRIS "People and Robots" (CPAR) Initiative. The authors were supported in part by the Scalable Collaborative Human-Robot Learning (SCHooL) Project, an NSF National Robotics Initiative Award, and by donations from Google and Toyota Research Institute and equipment grants from PhotoNeo, Nvidia, and Intuitive Surgical. Any opinions, findings, and conclusions or recommendations expressed in this material are those of the author(s) and do not necessarily reflect the views of the sponsors. We thank our colleagues who provided helpful feedback, especially Suraj Nair, Zaynah Javed and Daniel Brown. Ashwin Balakrishna was supported by an NSF GRFP.

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
