# OpenReview forum: "LS3: Latent Space Safe Sets for Long-Horizon Visuomotor Control of Sparse Reward Iterative Tasks"
_robot-learning.org/CoRL/2021/Conference — CoRL2021 Poster_

### Official Review · Reviewer_pzwD · 2021-07-22

**Originality:** Fair
**Technical Quality:** Good
**Clarity Of Presentation:** Good
**Impact:** 3

**Recommendation:**

Weak Accept: I recommend accepting the paper, but will not argue for my recommendation if the majority of other reviewers have a different opinion.

**Summary:**

This paper presents a visuomotor reinforcement learning algorithm (LS3) that attempts to guarantee safe learning and minimize constraint violations during the exploration process. It achieves this by leveraging an offline dataset containing some examples of task successes and some examples of constraint violations. A latent representation is first pre-trained (using a VAE) on the offline demonstrations, and then the algorithm uses model-predictive control (MPC) to learn several classifiers, a value function, and a latent dynamics model, to try and guarantee safe model-based exploration rollouts during the training process. The algorithm is shown to be effective across 3 simulated tasks and a physical cable routing task.

**Issues:**

Comment on the differences between SAVED and the proposed algorithm, especially if there are changes in this algorithm that are non-trivial, or that are critical for success in visuomotor settings compared to settings with low-dimensional ground-truth observations. If possible, include a baseline comparison against SAVED (as an upper bound on performance with image observations), and additional ablations of the method that bring it closer to SAVED -- these comparisons would be helpful in understanding the novelty of the work.

**Reviewer Expertise:**

Very good: Comprehensive knowledge of the area

**Strengths And Weaknesses:**

Strengths

The paper is mostly well-written (apart from a few clarity issues), and the problem it addresses is well-motivated. The margin by which the proposed algorithm (LS3) outperforms baselines in the physical robot experiments is also very impressive. The video overview is also very useful.

Weaknesses

One of the biggest issues is that this paper appears to be extremely similar to another algorithm (SAVED, reference 18 in the paper). The methodology of using safe sets during MPC for safe exploration was introduced in that paper. The main novelty in this paper appears to be extending that algorithm to visuomotor tasks. This is not necessarily a bad thing, but from a technical standpoint, the main difference appears to be pre-training a VAE on an offline dataset, and using the encoder to embed all image states into latent states -- all other components are then trained on latent states instead of image states.

If there are more important technical ingredients that are different from SAVED, they should be stated more explicitly, and ablations should be included to show their importance. Adding SAVED as an additional baseline (and providing it with access to to ground-truth low-dimensional observations) could be useful to provide an upper bound on performance.

There are also some clarity issues. In the title, the paper mentions two keywords - "long-horizon" and "iterative", but what they refer to is not very clear in the paper text. In section 5.3, it is stated that the horizon for these tasks are 50-150 timesteps - this does not appear to be long-horizon, as compared to prior work. Some point of comparison for why these tasks are long-horizon is necessary, especially when considering prior work in safe reinforcement learning. Similarly, the paper mentions "iterative learning control" where an agent usually tracks a reference trajectory, but what is an "iterative task"?

More detailed comments and questions follow:
- Is it necessary to update the VAE using online data? If not, why does it suffice to learn it purely on the offline dataset - is it because the amount of variation in start and goal states is small?
- lines 133-136 were confusing - make sure to state that learning approximations to R and C is okay, you just assume that you can't evaluate the R and C functions directly on future predicted states (hence the need for learned approximations)
- The definitions in lines 176-179 are pretty convoluted - it seems like an equivalent way to say the same thing is that the set $S^j$ contains the states visited by all successful trajectories in iteration j. The next lines futher complicate things - what's the difference between $D_{success}$ and $S^j$? Is $D_{success}$ just the union over all $S^j$?
- SACfD + RRL appears to be a very strong baseline, matching asymptotic performance on the two harder simulated tasks (reacher and sequential pushing). This is why, at first glance, there appeared to be a large discrepancy between Table 1 and Figure 4. The task success rate was much larger for LS3 than SACfd+RRL, but as previously mentioned, SACfD+RRL appears to match or exceed the return of LS3 on 2 of the 3 tasks. Lines 275-276 suggest that the table is reported over the entire training process, as opposed to the final trained policy - this is understandable for the constraint violation metric, but unconventional for task success rate. The table caption should be made more clear - perhaps change the bolded portion to read "Task Success Rate over all Training Episodes".
- Are lines 187-188 defining the relaxed Latent Safe Set via the normalized classifier logits (e.g. the level $\delta_S$ = 0.5 would correspond to normal classification)?
- Should equation 2 have a discount factor in front of $f_G$ (since the value function appears to correspond to discounted returns)? Furthermore, the preliminaries section suggests that negative rewards are used ({-1, 0}), but the classifier defines sparse positive rewards ({0, 1}) - which one is used?
- Equation 4 and 5 both use j, but in seemingly different contexts
- line 218 - typo ("how to the task")

**Summary Of Recommendation:**

While this paper presents interesting empirical results, it is hard to assess the value of the contribution without understanding more about how it differs to its nearest neighbor (SAVED, reference 18), and whether the proposed algorithms amounts to running SAVED with a pre-trained latent representation.

(Post-Discussion). The authors have addressed most of my comments, and I have changed my score accordingly

---

> ### Author Response · Authors · 2021-08-21
> **Response to Reviewer pzwD part 1**
>
> **We thank reviewer pzwD for their comments and address each comment below.  We have additionally attached an updated draft of the paper and supplement with all the changes indicated in the responses below highlighted in blue. We have split this response into multiple parts due to space constraints.**
>
> “Adding SAVED as an additional baseline (and providing it with access to ground-truth low-dimensional observations) could be useful to provide an upper bound on performance.”
>
> **We agree that this would be a valuable comparison, but we are not quite sure that this would necessarily provide an upper bound on performance since the fundamental mechanism by which the Safe Set operates is quite different between SAVED and LS3 as explained above. Additionally, due to the mechanism by which SAVED represents the Safe Set, computational experiments are very slow to run (as discussed in the previous responses), and may not be feasible by the rebuttal deadline. However, we still think this would be an interesting additional comparison which we will add to the paper in the camera ready version if accepted.**
>
> “In section 5.3, it is stated that the horizon for these tasks are 50-150 timesteps - this does not appear to be long-horizon, as compared to prior work. Some point of comparison for why these tasks are long-horizon is necessary, especially when considering prior work in safe reinforcement learning.”
>
> **A task horizon of 150 steps is quite long compared to prior work in image-based RL with sparse reward functions, which typically consider very short horizon control tasks [1, 5, 6] (< 20 steps).**
>
> “Similarly, the paper mentions "iterative learning control" where an agent usually tracks a reference trajectory, but what is an "iterative task"?”
>
> **Learning MPC presents a reference free iterative learning control algorithm where an agent constructs a Safe Set to iteratively improve upon an initial set of feasible but potentially suboptimal demonstrations in a task with low variance start state distribution and fixed goal set. We also consider this same general problem setting, but consider significantly more complex visuomotor control tasks. We define the iterative learning control setting (eg. iterative tasks) in Section 3, but have updated the wording to make this definition more clear.**
>
> “Is it necessary to update the VAE using online data? If not, why does it suffice to learn it purely on the offline dataset - is it because the amount of variation in start and goal states is small?”
>
> **Updating the VAE using online data may improve task performance but we found that the offline dataset used for LS3 and the other comparisons contained sufficient information to learn good quality embeddings without online data. We experimented with updating the VAE using online rollouts from the agent, but found that this led to a number of instabilities since this changed the domain of the various models defined on top of the representations learned by the VAE (dynamics model, Safe Set, goal and constraint indicators).**
>
> “lines 133-136 were confusing - make sure to state that learning approximations to R and C is okay, you just assume that you can't evaluate the R and C functions directly on future predicted states (hence the need for learned approximations)”
>
> **Sorry for the confusion. We have updated Section 3 in the paper accordingly to clarify that we assume that R and C may be approximated using prior data for use during planning.**
>
> “The definitions in lines 176-179 are pretty convoluted - it seems like an equivalent way to say the same thing is that the set Sj contains the states visited by all successful trajectories in iteration j. The next lines further complicate things - what's the difference between Dsuccess and Sj? Is Dsuccess just the union over all Sj?”
>
> **To clarify, Sj is the set of all states in successful trajectories up to and including iteration j where demonstrations are denoted as successful trajectories collected in iteration 0. Dsuccess and Sj are actually equivalent. We agree that this may have been confusing, and have simplified the writing and clarified the above in Section 4.2 accordingly.**

---

> > ### Author Response · Authors · 2021-08-21
> > **Response to Reviewer pzwD part 2**
> >
> > “SACfD + RRL appears to be a very strong baseline, matching asymptotic performance on the two harder simulated tasks (reacher and sequential pushing). This is why, at first glance, there appeared to be a large discrepancy between Table 1 and Figure 4. The task success rate was much larger for LS3 than SACfd+RRL, but as previously mentioned, SACfD+RRL appears to match or exceed the return of LS3 on 2 of the 3 tasks.”
> >
> > **The objective in this work is not to maximize asymptotic performance (for which all methods do well), but to investigate how the learned Safe Set can improve learning efficiency for sparse reward tasks. Accordingly, Figure 4 and Table 1 illustrate that LS3 is able to learn substantially more efficiently than prior algorithms by leveraging the learned Safe Set to structure exploration based on prior task successes, while comparisons which do not structure exploration in this way are significantly less sample efficient.**
> >
> > “ Lines 275-276 suggest that the table is reported over the entire training process, as opposed to the final trained policy - this is understandable for the constraint violation metric, but unconventional for task success rate. The table caption should be made more clear - perhaps change the bolded portion to read "Task Success Rate over all Training Episodes".”
> >
> > **Agreed and sorry for the confusion, we have updated the Table caption and the bolded portion as suggested.**
> >
> > “Are lines 187-188 defining the relaxed Latent Safe Set via the normalized classifier logits (e.g. the level δS= 0.5 would correspond to normal classification)?”
> >
> > **deltaS=0.5 would correspond to classifying a state as part of the safeset if the estimated probability of inclusion is at least 0.5. However, the recursive formulation of our latent Safe Set classifier as mentioned in lines 186-188 additionally incorporates temporal structure in the trajectories and encourages consecutive states in trajectories to have similar values under the learned classifier.**
> >
> > “Should equation 2 have a discount factor in front of fG (since the value function appears to correspond to discounted returns)? Furthermore, the preliminaries section suggests that negative rewards are used ({-1, 0}), but the classifier defines sparse positive rewards ({0, 1}) - which one is used?”
> >
> > **To clarify, we do not use discounted returns in this work, so no discount factor should appear. Sorry for the confusion in the definition of rewards, we used negative sparse rewards  ({-1, 0} for all experiments), and the classifier defined in Section 4.3 is trained to estimate whether a state is in the goal set or not (hence the {0, 1} labels).**
> >
> > “Equation 4 and 5 both use j, but in seemingly different contexts”
> >
> > **You are correct, sorry for the confusion. To keep notation consistent with Section 4.2 and the rest of the paper, we have changed the j in equation 4 to k.**
> >
> > “line 218 - typo ("how to the task")”
> >
> > **Thanks for catching this, we have updated this to “how to do the task”**

---

> > > ### Author Response · Authors · 2021-08-21
> > > **Response to Reviewer pzwD part 3**
> > >
> > > “One of the biggest issues is that this paper appears to be extremely similar to another algorithm (SAVED, reference 18 in the paper). If there are more important technical ingredients that are different from SAVED, they should be stated more explicitly, and ablations should be included to show their importance.”
> > >
> > > **While the LS3 algorithm is inspired by theoretically sound control algorithms such as SAVED [16] and learning MPC [14], successfully extending these ideas to visuomotor control required a number of algorithmic contributions. First, while Safe Sets have been studied widely in prior work in control theory [14, 15, 16, 17, 20] , using them in practice for control requires constructing continuous approximations to these sets. Thus, the key contribution of LS3 is in designing a continuous approximation method which is significantly more scalable to high dimensional observations than those used in prior work. Second, while SAVED and prior LMPC algorithms leverage access to ground truth state during planning to determine whether states are constraint violating or in the goal set, predicted images typically do not expose the necessary information to exactly determine these properties. Thus, the second key difference in LS3 is in the need to infer whether predicted images are in the goal set or constraint violating from prior data. We apologize that the algorithmic distinctions between LS3 and SAVED were not stated more explicitly in the Abstract and Introduction (Section 1) sections and we have updated these sections to emphasize this. We discuss the first point (the key contribution) in more detail below.**
> > >
> > > **While Safe Sets have been studied widely in control theory literature in the past [14, 15, 16, 17, 20] and can be used to design algorithms with a number of desirable theoretical properties, their representation has a critical impact on the problems that these algorithms can be applied to. Rosolia et. al [14] introduced the definition of the sampled Safe Set (set of all states from which the task was previously successfully completed) which we use in this work. To operationalize this Safe Set for planning, they then define a continuous approximation by taking the convex hull of all states in this sampled Safe Set. While this choice leads to a number of desirable theoretical properties given known dynamics which satisfy certain regularity conditions, approximating the Safe Set with a convex hull not only significantly limits the space of continuous sets which can be represented, but also results in a linear time lookup to determine whether a state lies in this continuous Safe Set. SAVED [16] introduces a new method to represent the Safe Set using Kernel Density estimation with a nearest neighbors kernel. This allows the Safe Set to more easily represent significantly more complex continuous sets in the state space, enabling scalability to more challenging robotic tasks. However, as noted in the SAVED paper, kernel density estimation does not scale effectively to high dimensions, and correspondingly, all tasks considered in the paper have a state space dimension of less than 20. Furthermore, kernel density estimation with a nearest neighbors kernel also has a linear time lookup, making it infeasible to use this Safe Set when learning from large datasets, which is almost always required when learning robotic tasks from visuomotor inputs.**
> > >
> > > **In this work, we introduce a new continuous approximation for Safe Sets which for the first time makes it possible to apply the insights from the learning MPC class of algorithms to high-dimensional robotic control tasks. Our first insight is that, given examples of unsuccessful trajectories, the Safe Set can instead be represented as the super level set of a binary classifier, which estimates the probability of a given state being an element of a successful trajectory. Our second insight is that while learning a binary classifier is appealing for its simplicity and scalability, it does not explicitly leverage the temporal structure in successful trajectories. To address this, we train a binary classifier with a recursive objective to predict whether a state is an element of a successful trajectory or whether the next state in the trajectory is likely to be in a successful trajectory. This captures the intuition that if some state s_{t+1} is safe, it is likely that s_{t} is also safe. We find that this continuous approximation method scales gracefully to high dimensional observations while yielding strong performance in practice. We have additionally added an ablation studying the importance of training the Safe Set classifier with the above recursive objective (see updated Figure 4) and find that particularly in the challenging Sequential Pushing environment, this recursive objective is critical for LS3’s performance.**

---

> > > > ### Comment · Reviewer_pzwD · 2021-08-23
> > > > **Response**
> > > >
> > > > Thank you for addressing my comments. I appreciate the clarification between SAVED and the proposed method, and the fact that SAVED comparison would be added to the paper if accepted. Regarding my point about what makes these tasks "long-horizon", I would further clarify in the text that the key distinction is that these tasks have sparse rewards (separating them from other longer horizon pixel-based RL works that use dense rewards) - this was not clear to me in the text. I will consider increasing my score, pending further reviewer discussion with the other reviewers

---

> > > > > ### Author Response · Authors · 2021-08-24
> > > > > **Thank you for your comments, Paper has been updated to emphasize Sparse Rewards**
> > > > >
> > > > > We do discuss that most prior work in model-based RL typically assumes dense cost functions in the related work section, but agree that it would be helpful to further emphasize that we consider sparse reward tasks in this work, and that this is a differentiating factor that makes the problem setting more challenging. Accordingly, we have updated (see new draft) the paper title, Introduction, and Evaluation Metrics subsection of the Experiments to emphasize that the tasks we consider have sparse rewards (motivated by the fact that dense reward specification is often very difficult from image inputs), making efficient and safe exploration especially challenging.
> > > > >
> > > > > We thank you for the positive feedback and careful consideration of our response. Please let us know if any further clarification would be helpful in better understanding the paper and its contributions.

---

### Official Review · Reviewer_5joi · 2021-07-23

**Originality:** Very Good
**Technical Quality:** Very Good
**Clarity Of Presentation:** Excellent
**Impact:** 4

**Recommendation:**

Weak Accept: I recommend accepting the paper, but will not argue for my recommendation if the majority of other reviewers have a different opinion.

**Summary:**

The paper presents an approach for exploration in image-based tasks in the presence of unsafe regions, i.e. when exploration is constrained. The algorithm, LS3, learns a latent space which enforces reconstruction, dynamics, a value function, and most novelty and crucially a safe set. This safe set is constructed from demonstration states which previously resulted in success and is used to guide its exploration towards safe regions. LS3 is demonstrated on several robotic tasks to demonstrate it learns faster than other state of the art methods and at times to higher performance.

**Issues:**

There are two issues that I believe are important to investigate to fully understand the performance of LS3:
- One potential drawback of this method is the converged performance due to limitations in exploration to safe regions. Figure 4 cuts off training at 250 trajectories. It should be shown on a log scale to a larger number of trajectories.
- Table 2 shows that LS3 actually violates constraints more often than other methods in some domains. As LS3 is particularly created to allow safe exploration, this is a surprising result. However it is only briefly discussed and conjectured about. Though the hypothesis seems reasonable, it should be verified and thoroughly understood since it is somewhat antithetical to the purpose of this work.

Minor notes:
- An interesting avenue that is not discussed is how the approach performs with more complex dynamics, such as systems with velocity and dynamical constraints. These are common systems that require safety and staying in safe regions. This however complicates the state space construction and may make the learning much more difficult. It would be interesting to understand the potential of this method in those settings.
- Figure 2 should say where each of the different learned sections are discussed.
- The on robot task is appreciated, but I believe the result would be much stronger with a pushing task.


**Reviewer Expertise:**

Good: General knowledge of the area

**Strengths And Weaknesses:**

Strengths:
- The paper is clear and interesting
- The choices for network architecture are clear and reasonable
- The approach learns quickly and safely over a range of tasks and on robot experiments

Weaknesses:
- It is not clear how much exploration is limited compared to other approaches
- The constraint violation rate of other algorithms is surprising given it is a focus of the approach and should be investigated more


**Summary Of Recommendation:**

The algorithm is well motivated and clearly described in this work. The results are promising and demonstrated on a number of tasks, including an on robot experiment. A few results require deeper investigation, particularly in how this approach limits exploration in the limit and surprising results on safety of exploration during learning.

---

> ### Author Response · Authors · 2021-08-21
> **Response to Reviewer 5joi**
>
> **We thank reviewer 5joi for their comments and address each comment below.  We have additionally attached an updated draft of the paper and supplement with all the changes indicated in the responses below highlighted in blue.**
>
> “It is not clear how much exploration is limited compared to other approaches”
>
> **The key function of the Safe Set in LS3 is to help restrict exploration to the neighborhood of prior task successes, allowing for more efficient exploration. This is illustrated in the learning curves in Figure 4 and the success rates in Table 1, which illustrate that LS3 explores substantially more efficiently than an ablation without the Safe Set, indicating that the Safe Set makes it possible to effectively restrict exploration to rapidly improve performance and consistently succeed at the task during learning.**
>
> “The constraint violation rate of other algorithms is surprising given it is a focus of the approach and should be investigated more”
>
> **We apologize for any miscommunication of the results. We do believe that the safety results are empirically compelling, because LS3 is the most safe algorithm considered that makes significant task progress, while other algorithms are safer because they do not make much task progress.**
>
> “One potential drawback of this method is the converged performance due to limitations in exploration to safe regions. Figure 4 cuts off training at 250 trajectories. It should be shown on a log scale to a larger number of trajectories.
>
> **As LS3 is a model-based RL algorithm, executing LS3 for a large number of training iterations is computationally expensive due to the stochastic sampling based optimization involved in planning, which is also commonly in recent model-based RL literature [6, 25, 26]. Additionally, LS3 had converged by the time that the plots were cut off, so we are not sure if more training iterations would provide much additional information.**
>
> “Table 2 shows that LS3 actually violates constraints more often than other methods in some domains. As LS3 is particularly created to allow safe exploration, this is a surprising result. However it is only briefly discussed and conjectured about. Though the hypothesis seems reasonable, it should be verified and thoroughly understood since it is somewhat antithetical to the purpose of this work.”
>
> **Prior algorithms we compare to make insignificant or slow learning progress during the recorded training iterations, which enable them to sometimes violate constraints less than LS3. In the process of learning to perform the task, LS3 spends much more time in regions where constraint violations are possible. This is evidenced by the stark disparity between task success rates of SACfD and LS3 in Table 1. In one extreme, a policy that learns to stay still or doesn’t explore at all will also have few constraint violations. LS3 explores and learns to perform the task quickly, and still has relatively few constraint violations. This difference is particularly apparent in the physical experiment, where prior algorithms are unable to effectively explore while LS3 is able to rapidly learn the task. We have updated the writing in Section 5 to make this point more clear.**
>
> “An interesting avenue that is not discussed is how the approach performs with more complex dynamics, such as systems with velocity and dynamical constraints. These are common systems that require safety and staying in safe regions. This however complicates the state space construction and may make the learning much more difficult. It would be interesting to understand the potential of this method in those settings.”
>
> **We fully agree that this would be an exciting avenue for future work and we have updated Section 6 to discuss these points accordingly.**
>
> “Figure 2 should say where each of the different learned sections are discussed.”
>
> **The Figure 2 caption does indicate the sections in which each of the learned functions are discussed (Sections 4.1-4.3). We are happy to add more details to the caption, but would appreciate any clarification on what should be added.**
>
> “The on robot task is appreciated, but I believe the result would be much stronger with a pushing task.”
>
> **We agree that a pushing task would be interesting on the robot as well, and are excited to try this in future work. We explore cable manipulation due to the challenges of analytically modeling cable deformations, which strongly motivate learning visual dynamics models.**

---

### Official Review · Reviewer_phDr · 2021-07-23

**Originality:** Fair
**Technical Quality:** Fair
**Clarity Of Presentation:** Good
**Impact:** 3

**Recommendation:**

Weak Reject: I recommend rejecting the paper, but will not argue for my recommendation if the majority of other reviewers have a different opinion.

**Summary:**

The paper proposes a framework for safe learning with visual inputs for iterative tasks. The paper combines existing work on learning latent dynamics from visual input and learning-based model predictive control. In particular, the proposed method learns a safe set in the latent space as a terminal constraint for the MPC algorithm.

**Issues:**

See the "Strengths and Weaknesses" part.

**Reviewer Expertise:**

Very good: Comprehensive knowledge of the area

**Strengths And Weaknesses:**

Strengths:
- The paper considers an important problem of safe learning with visual inputs, which has great potential impact.
- The figures in the paper are informative and helpful for readers to understand the paper.
- I appreciate the level of details of the algorithm implementation provided in the appendices.
- The authors conduct thorough sensitivity study on the safety parameters for the proposed algorithm as well as the baselines.

Weaknesses:
- In terms of novelty, this work also seems to mostly a combination of learning latent visual dynamics [2] and SAVED [18].
- Although the primary claims of the paper seems to be "ensure safety" (see line 53-56), this is unfortunately not sufficiently supported by the rest of the paper, both theoretically and experimentally. Personally I don't think the authors are required to address safety *both* theoretically and experimentally. However, it would be good if the authors can justify either one of them during the rebuttal period or in the revision.
  - In the theoretical point of view, learning-based MPC [19] only provides safety and feasibility guarantees when 1) the ground truth dynamics is used within the MPC and 2) the dynamics is deterministic. Both of them don't hold in this paper, as learned stochastic dynamics are used. Even if we assume that the learned dynamics is accurate, the fact that it is stochastic already breaks a lot of the premises of learning-based MPC. As an example, the latent space safe set actually does not "ensures that the agent can plan back to regions in which it is conﬁdent in task completion" (as the authors claimed in line 53), because having one success example for a stochastic MDP doesn't ensure that we can return to the safe region even if we apply the same actions.
  - Algorithmically, I see that the proposed method makes some compromises because stochastic learned dynamics breaks the assumptions in learning-based MPC. These compromises can provide us a practical algorithm (in the sense that we can run them), but could be even more harmful to safety. For example, the authors solve the MPC problem, which is a constrained optimization problem, using CEM. However, it is possible that all particles violate the constraints of terminating in the safe set, which is noted by the authors in the appendices on choosing $\delta_S$ (another cause of this is the broken of recursive feasibility due to stochastic dynamics). To mitigate this, the authors proposed to online adjust $\delta_S$ such that there exists a constraint satisfying particle. The experimental results seem to show that the performance with and without the safe set are quite similar except that the policy starts with a lower reward without the safe set constraint. I hope the authors can further clarify why this performance gap at the beginning of training is important.
  - In terms of the experimental results, the proposed method does not show significant improvement over baselines in terms of constraint violations. Much to my surprise, SACfD, which is not even a safe RL approach, does almost as well as the proposed method in the point mass environment, only 1 std worse than the proposed method in the reacher task, and significantly better in the sequential pushing task. Although the authors explained that this is partially due to that SACfD also has worse performance. I still find it a little unsatisfying as SACfD is performing almost as well as the proposed algorithm at the end of learning in terms of the reward.
- I find it a little hard to interpret the results from the cable routing task. I hope the authors can further explain why this task is challenging. The robot is controlled by $\Delta x$ and $\Delta y$ in the end-effector space and the robot is moving quite slowly according to the videos. It seems to me that one can almost achieve the task by replaying successful demonstrations, or just following an end-effector trajectory given by some simple heuristics, e.g. moving straight enough then take a turn. This makes me wonder why RL is a good approach to solve this problem. I do like to know the authors' thoughts on this.
- I also hope that the authors can clarify how function $f_S$ is learned. In line 186-187, the authors seem to define $f_S$ as a solution to a fixed point problem. However, in the appendices, the authors seem to just treat learning $f_S$ as a classification problem. It would be great if the authors can provide more information on this.

**Summary Of Recommendation:**

The paper seeks to solve the problem of safe RL with visual input, which is an important research question. However, the paper seems to be more of a combination of existing work in learning visual dynamics and learning-based MPC. The combination also breaks some of the sound theoretical properties of learning-based MPC, as I mentioned in the "Strengths and Weaknesses" part. The experimental results at this point were not sufficient to justify the contribution of safe learning as standard RL algorithm (SAC) seems to do as well as the proposed method in terms of safety and reward. I hope that the authors can provide more justifications on the theoretical soundness of the paper and/or the experimental results. I am happy to update my reviews during later stages.

---

> ### Author Response · Authors · 2021-08-21
> **Response to Reviewer phDr part 1**
>
> **We thank reviewer phDr for their comments and address each comment below.  We have additionally attached an updated draft of the paper and supplement with all the changes indicated in the responses below highlighted in blue. We have split this response into multiple parts due to space constraints.**
>
> “Although the primary claims of the paper seems to be "ensure safety" (see line 53-56), this is unfortunately not sufficiently supported by the rest of the paper, both theoretically and experimentally. Personally I don't think the authors are required to address safety both theoretically and experimentally. However, it would be good if the authors can justify either one of them during the rebuttal period or in the revision.”
>
> **We believe that there is a significant miscommunication of the results, which is likely our fault. We do believe that the safety results are empirically compelling, because LS3 is the most safe algorithm considered that makes significant task progress, while other algorithms are safer because they do not make much task progress. We will address this point below. However, due to all of the practical assumptions, we do wish to de-emphasize safety as a contribution, and focus on the scalability of the method to high dimensional dynamical systems, which leads to efficient learning compared to baselines that do not use the new Safe Set formulation. We have clarified our discussion of the results in Section 5 to reflect the above and have changed the phrasing in the paper from “ensure safety” from the paper to “encourage safety”.**
>
> “In the theoretical point of view, learning-based MPC [19] only provides safety and feasibility guarantees when 1) the ground truth dynamics is used within the MPC and 2) the dynamics is deterministic. Both of them don't hold in this paper, as learned stochastic dynamics are used. Even if we assume that the learned dynamics is accurate, the fact that it is stochastic already breaks a lot of the premises of learning-based MPC. As an example, the latent space Safe Set actually does not "ensures that the agent can plan back to regions in which it is conﬁdent in task completion" (as the authors claimed in line 53), because having one success example for a stochastic MDP doesn't ensure that we can return to the safe region even if we apply the same actions.”
>
> **There is significant recent work on designing learning-based MPC algorithms with safety guarantees for stochastic, and sometimes even nonlinear systems [15, 17]. The key, as the reviewer pointed out, is not reasoning about individual trajectories, but sequences of robust reachable sets, which can be approximated in practice by sampling many trajectories from the same controllers. Nonetheless, because we don’t provide safety guarantees under the practical assumptions we made in this work to increase scalability to high dimensional image inputs, we will de-emphasize safety and instead focus our presentation on the novel method to learn “Safe Sets” in high dimensional spaces.**
>
> “For example, the authors solve the MPC problem, which is a constrained optimization problem, using CEM. However, it is possible that all particles violate the constraints of terminating in the Safe Set, which is noted by the authors in the appendices on choosing δS”
> “ (another cause of this is the broken of recursive feasibility due to stochastic dynamics). To mitigate this, the authors proposed to online adjust δS such that there exists a constraint satisfying particle. The experimental results seem to show that the performance with and without the Safe Set are quite similar except that the policy starts with a lower reward without the Safe Set constraint. I hope the authors can further clarify why this performance gap at the beginning of training is important.”
>
> **Because of these practical assumptions, we have de-emphasized safety as a core contribution of the algorithm in the Abstract and Introduction. However, the benefits of the Safe Set in classical literature are two-fold: safety and also efficient learning by constraining exploration of the agent to regions where the agent has performed well in the past. As the reviewer points out, LS3 learns more efficiently with the Safe Set than without it, suggesting that its new Safe Set formulation can also increase learning efficiency.**

---

> > ### Author Response · Authors · 2021-08-21
> > **Response to Reviewer phDr part 2**
> >
> > “In terms of the experimental results, the proposed method does not show significant improvement over baselines in terms of constraint violations. Much to my surprise, SACfD, which is not even a safe RL approach, does almost as well as the proposed method in the point mass environment, only 1 std worse than the proposed method in the reacher task, and significantly better in the sequential pushing task. Although the authors explained that this is partially due to that SACfD also has worse performance. I still find it a little unsatisfying as SACfD is performing almost as well as the proposed algorithm at the end of learning in terms of the reward.”
> >
> > **SACfD makes insignificant/slow learning progress during the recorded training iterations, which is why it violates constraints the least. In the process of learning to perform the task, LS3 spends much more time in regions where constraint violations are possible. This is evidenced by the stark disparity between task success rates of SACfD and LS3 in Table 1. In one extreme, a policy that learns to stay still or doesn’t explore at all will also have few constraint violations. LS3 explores and learns to perform the task quickly, and still has relatively few constraint violations. This difference is particularly apparent in the physical experiment, where prior algorithms are unable to effectively explore while LS3 is able to rapidly learn the task.**
> >
> > “I find it a little hard to interpret the results from the cable routing task. I hope the authors can further explain why this task is challenging. The robot is controlled by Δx and Δy in the end-effector space and the robot is moving quite slowly according to the videos. It seems to me that one can almost achieve the task by replaying successful demonstrations, or just following an end-effector trajectory given by some simple heuristics, e.g. moving straight enough then take a turn. This makes me wonder why RL is a good approach to solve this problem. I do like to know the authors' thoughts on this.”
> >
> > **While the motions of the surgical robot are artificially slowed down to prevent damage to the robot, we measure robot performance in terms of timesteps, or number of actions taken to reach the goal set. The learner is able to significantly outperform the demonstrator, and it is challenging to hard-code heuristic-based methods to perform this task well, as the heuristic would need to account for complex cable deformations during manipulation in order to avoid hitting the obstacle in the scene. This motivates a learning-based technique that can optimize the robot’s trajectory by learning how actions deform the cable.**
> >
> > “I also hope that the authors can clarify how function fS is learned. In line 186-187, the authors seem to define fS as a solution to a fixed point problem. However, in the appendices, the authors seem to just treat learning fS as a classification problem. It would be great if the authors can provide more information on this.”
> >
> > **Thanks for pointing this out. While lines 186-187 do not contradict the appendix, we have updated the appendix to further describe how the fixed point iteration problem is solved. Namely, the classifier is iteratively trained on target probabilities that are partially generated by evaluating itself on subsequent states. Then in the next iteration of training, the targets are regenerated on the next iterate of the classifier and a classification loss is minimized. This is very similar to how value functions are trained to solve a fixed point iteration problem in deep reinforcement learning.**

---

> > > ### Author Response · Authors · 2021-08-21
> > > **Response to Reviewer phDr part 3**
> > >
> > > “However, the paper seems to be more of a combination of existing work in learning visual dynamics and learning-based MPC. The combination also breaks some of the sound theoretical properties of learning-based MPC, as I mentioned in the "Strengths and Weaknesses" part.”
> > >
> > > **As discussed before, we will de-emphasize safety, as we do not provide safety guarantees, but as discussed above and below, we do note that LS3 is significantly safer than algorithms that also efficiently learn to perform the tasks (Tables 1 and 2), and we have emphasized this in the text in blue.**
> > >
> > > **We also note that although LS3 does combine some ideas from learning-based MPC and visuomotor control, the key innovation in LS3 is in the method of creating a continuous approximation for the Safe Set so that it can be used for visuomotor control. Prior works introduce options for this continuous approximation such as constructing a convex hull of the sampled Safe Set [14] or using kernel density estimation with a tophat kernel [16]. However, these methods have a number of scalability challenges which make them difficult to use for control with high-dimensional observations. In this work, we introduce a new continuous approximation method for the Safe Set by reframing the Safe Set approximation as a binary classification problem in a learned latent space, where the objective is to distinguish states from prior successful trajectories from those in prior unsuccessful trajectories. We expand on these points in our response to reviewer pzwD and have edited the Abstract and Introduction to emphasize these contributions accordingly.**
> > >
> > > “The experimental results at this point were not sufficient to justify the contribution of safe learning as standard RL algorithm (SAC) seems to do as well as the proposed method in terms of safety and reward. I hope that the authors can provide more justifications on the theoretical soundness of the paper and/or the experimental results.”
> > >
> > > **As stated above, SAC has few constraint violations during learning, because it does not explore efficiently enough to even reach constraint violating states in early episodes, let alone perform the task. This can be quantified by viewing its low task success rates in Table 1, whereas LS3 has high success rates in Table 1, suggesting that it efficiently learned to do the tasks, and relatively low constraint violations in Table 2, when compared to methods that are also able to learn the tasks. We have updated Section 5 to clarify this point further in writing, and we hope the reviewer will reconsider this point, as we believe it is a significant misunderstanding of the results (likely our fault).**

---

### Official Review · Reviewer_aHEL · 2021-07-24

**Originality:** Very Good
**Technical Quality:** Very Good
**Clarity Of Presentation:** Very Good
**Impact:** 4

**Recommendation:**

Strong Accept: I recommend accepting the paper and will argue for my recommendation even if other reviewers hold a different opinion.

**Summary:**

The paper proposes the Latent space Safe Sets LS3 method aiming solve the vision-based control tasks safely and more efficiently. The approach is based on modeling latent probabilistic model of the world, classifying future constraint violation, which enables trajectory optimization with estimated probability of constraint violation over this latent probabilistic dynamical model.

A key assumption is that a dataset containing both colliding and free transitions in the environment is provided. Learning proceeds by first training the world dynamical model, variational decoder, encoder, safe set classified, value function goal indicator, and constraint estimator, and then collecting data during execution and updating the model with new data.

Simulation experiments on 3 vision-based control tasks and real-world experiment with a cable-routing task on a daVinci robot are presented.



**Issues:**

Please address the list of the weaknesses above, as much as possible.

**Reviewer Expertise:**

Very good: Comprehensive knowledge of the area

**Strengths And Weaknesses:**

Strengths:
+ relies on powerful latent-space world-state representation with VAE to encode uncertainty
+ a unified framework for predicting the world state and incorporating constraints in an efficient manner (through reduced latent space)
+ use of trajectory selection/optimization with constraints can give more confidence than e.g. a learned policy
+ the simulation studies are interesting and a nice illustration of the method

Weaknesses:
- details of the CEM optimizer are missing (even in the references these details are hard to find), i.e. would be interesting to say how these probabilities are estimated, if it is from samples how many samples are needed, is there a way to come up with a measure of trust in this estimation since this is what the original objective of the paper is...
- the sample-based optimization relies on stochastic rollouts and it is not super clear what the role of uncertainty is vs samples to perturb the actions in CEM
- shouldn't the recurrent dynamics block also encode uncertainty?

**Summary Of Recommendation:**

The paper presents a principled way to incorporate safety, and while some key details about how probability distributions are approximated is missing, it is an important and promising contribution.

---

> ### Author Response · Authors · 2021-08-21
> **Response to Reviewer aHEL**
>
> **We thank reviewer aHEL for their comments and address each comment below.  We have additionally attached an updated draft of the paper and supplement with all the changes indicated in the responses below highlighted in blue.**
>
> “details of the CEM optimizer are missing (even in the references these details are hard to find), i.e. would be interesting to say how these probabilities are estimated, if it is from samples how many samples are needed, is there a way to come up with a measure of trust in this estimation?”:
>
> **These details are found in the section 7.1.1 in the appendix: including the class of distributions considered, number of samples, and number of iterations. The specific hyperparameters for CEM for each environment are found in Table 3 of the appendix. The details of this optimizer are extremely similar to prior work on planning with learned stochastic dynamics models [25, 26], but we will definitely provide additional implementation details, and also release the code implementation.**
>
> “the sample-based optimization relies on stochastic rollouts and it is not super clear what the role of uncertainty is vs samples to perturb the actions in CEM”
>
> **The optimization procedure has two separate sources of randomness, which are also used separately. For the same action sequence, samples from the learned stochastic dynamics model will have variation, and this variation is used to estimate the probabilities in equations 5 and 6, as discussed in lines 216 and 217. The second source of uncertainty is the perturbation of actions to iteratively optimize the action sequences during CEM. This perturbation enables CEM to try new action sequences, evaluate them, and then bias the sampling of new action sequences toward the highest performing ones in the previous iteration of CEM. The mechanics of CEM are explained in lines 501-517 in the appendix.**
>
> “shouldn't the recurrent dynamics block also encode uncertainty?”
>
> **We are not using a recurrent dynamics model in this work, though that would be an interesting avenue for future work. The learned dynamics encode uncertainty by leveraging the strategy from PETS [25]. Thus, we learn an ensemble of neural networks to represent the system dynamics, each of which output the parameters of a conditional Gaussian distribution over next states given the current state and action. This makes it possible to capture both aleatoric uncertainty due to inherent stochasticity in the dynamics, and epistemic uncertainty due to lack of data (via ensembling).**

---

### Meta-Review · Area_Chair_9zqk · 2021-08-09

**Recommendation:** Accept (Poster)
**Confidence:** 4

**Metareview:**

The paper addresses an important research challenge. Safety when learning from visual input is an important unsolved problem which is highly relevant for the robotics and machine learning communities. The paper is well written and technically sound. The paper mostly relies on combining existing techniques to achieve the desired end result.

Pros:
- Paper addresses an important hard problem
- Real robot experiments

Cons:
- To correspond to the claims, safety should be explicitly shown in theory or/and experimentally. Please, see comments from reviewers phDr, pzwD
- Experiments do not show an advantage over previous (non-safety targeted) methods. The paper should discuss this in more detail and potentially provide more experimental results to yield more insight into the benefits of the approach.

Update: The safety aspect has been clarified during the discussion period and the paper has been updated. The proposed approach does not provide theoretical safety guarantees but more a practical approach that uses a safety set to make exploration less susceptible to failures. There is experimental evidence of improved sample efficiency compared to comparison methods.

---

> ### Author Response · Authors · 2021-08-21
> **Response to Area Chair 9zqk**
>
> **We thank the meta-reviewer for their comments and address each comment below. We have additionally attached an updated draft of the paper and supplement with all the changes indicated in the responses below highlighted in blue.**
>
> “The paper mostly relies on combining existing techniques to achieve the desired end result.”
>
> **We agree in retrospect that we need to clarify the novel technical contributions in the paper.  We have updated the Introduction and Abstract to do so. In summary, while Safe Sets have been studied widely in control theory literature in the past [14, 15, 16, 17, 20] and can be used to design algorithms with a number of desirable theoretical properties, their representation has a critical impact on the problems that these algorithms can be applied to. Prior work introduced continuous approximation methods based on convex hulls [14] and kernel density estimation with a tophat kernel [16], but these methods do not effectively scale to high-dimensional observations.**
>
> **In this work, we introduce a new continuous approximation for Safe Sets which for the first time makes it possible to apply the insights from the learning MPC class of algorithms to high-dimensional robotic control tasks. Our first insight is that, given examples of unsuccessful trajectories, the Safe Set can instead be represented as the super level set of a binary classifier, which estimates the probability of a given state being an element of a successful trajectory. Our second insight is that while learning a binary classifier is appealing for its simplicity and scalability, it does not explicitly leverage the temporal structure in successful trajectories. To address this, we train a binary classifier with a recursive objective to predict whether a state is an element of a successful trajectory or whether the next state in the trajectory is likely to be in a successful trajectory. This captures the intuition that if some state s_{t+1} is safe, it is likely that s_{t} is also safe. We find that this continuous approximation method scales gracefully to high dimensional observations while yielding strong performance in practice. We have additionally added an ablation studying the importance of training the Safe Set classifier with the above recursive objective (see updated Figure 4) and find that particularly in the challenging Sequential Pushing environment, this recursive objective is critical for LS3’s performance. We expand on these points in our response to Reviewer pzwD.**
>
> “Safety should be explicitly shown in theory or/and experimentally”
>
> **The focus of this paper is to develop a practical, image-based algorithm inspired by theoretically sound control algorithms. Thus, we do not claim any theoretical contribution in this paper, but rather focus primarily on leveraging insights from theoretically sound ideas to develop a scalable algorithm which works well in practice. We have edited the Introduction section to omit any suggestions of explicit theoretical contribution for LS3 accordingly and provide detailed responses to the related comments from reviewers phDr, pzwD below.**
>
> “Experiments do not show an advantage over previous (non-safety targeted) methods. The paper should discuss this in more detail and potentially provide more experimental results to yield more insight into the benefits of the approach.”
>
> **We believe that the experiments do show a strong advantage of LS3 over prior work, but agree that we could have conveyed this advantage more clearly. We believe that Reviewer phDr may have significantly misread the results due to our potential miscommunication. The key benefit of the Safe Set in LS3 is the ability to restrict exploration to the neighborhood of prior successes, enabling consistent and sample efficient learning. Both Table 1 and the learning curves in Figure 4 clearly illustrate that LS3 is able to learn substantially faster than prior algorithms, and is able to achieve task successes more reliably and earlier than comparisons. Regarding constraint violations, a key property of all the environments we evaluate on is that constraint violations are much more likely when you are closer to effectively doing the task. For example, for the sequential pushing environment, an agent which is able to push the blocks towards the target location at the edge of the table will be significantly more likely to experience constraint violations where it pushes the blocks off the table until it has learned to avoid these violations. By contrast, an agent which rarely gets close to performing the task may not even interact with the blocks consistently, resulting in very few constraint violations. We have clarified our discussion of the results in Section 5 to reflect the above.**

---

> > ### Author Response · Authors · 2021-08-29
> > **Thank you for the valuable feedback!**
> >
> > **We deeply thank the reviewers and Area chair for their input during the rebuttal period, which we believe substantially improved both the clarity and technical contribution of the paper. We believe we have addressed all reviewer comments and illustrated that the proposed approach is (1) technically novel due to a new and scalable way to represent Safe Sets for safe and efficient model-based RL and (2) empirically safer and more efficient than baselines on an array of challenging, long-horizon, image-based domains, including a physical experiment on a cable routing task in which LS3 rapidly and safely learns the task while prior algorithms make little to no progress.**
> >
> > **To add further experiment support to (1), during the rebuttal period we added an ablation to the paper (see new Figure 4, LS3 (BC SS) experiment) which shows that the new recursive classification objective we introduce for learning the Safe Set is indeed critical for performance, further validating the algorithmic contributions of the paper. To add further experimental support to (2), we added a new plot in the supplement (see new Figure 1) which indicates that not only does LS3 achieve high rewards more quickly than prior algorithms, but also that LS3 is more reliably able to perform tasks at essentially all stages of training compared to prior algorithms.**

---

### Decision · Program_Chairs · 2021-09-13

**Decision:**

Accept (Poster)

**Comment:**

The paper addresses an important research challenge. Safety when learning from visual input is an important unsolved problem which is highly relevant for the robotics and machine learning communities. The paper is well written and technically sound. The paper mostly relies on combining existing techniques to achieve the desired end result.

Pros:
- Paper addresses an important hard problem
- Real robot experiments

Cons:
- To correspond to the claims, safety should be explicitly shown in theory or/and experimentally. Please, see comments from reviewers phDr, pzwD
- Experiments do not show an advantage over previous (non-safety targeted) methods. The paper should discuss this in more detail and potentially provide more experimental results to yield more insight into the benefits of the approach.

Update: The safety aspect has been clarified during the discussion period and the paper has been updated. The proposed approach does not provide theoretical safety guarantees but more a practical approach that uses a safety set to make exploration less susceptible to failures. There is experimental evidence of improved sample efficiency compared to comparison methods.